# Structure and Optical Anisotropy of Spider Scales and Silk: The Use of Chromaticity and Azimuth Colors to Optically Characterize Complex Biological Structures

**DOI:** 10.3390/nano13121894

**Published:** 2023-06-20

**Authors:** Denver Linklater, Arturas Vailionis, Meguya Ryu, Shuji Kamegaki, Junko Morikawa, Haoran Mu, Daniel Smith, Pegah Maasoumi, Rohan Ford, Tomas Katkus, Sean Blamires, Toshiaki Kondo, Yoshiaki Nishijima, Daniel Moraru, Michael Shribak, Andrea O’Connor, Elena P. Ivanova, Soon Hock Ng, Hideki Masuda, Saulius Juodkazis

**Affiliations:** 1Department of Biomedical Engineering, Melbourne University, Parkville, VIC 3010, Australia; denver.styczynski@unimelb.edu.au (D.L.); a.oconnor@unimelb.edu.au (A.O.); 2Stanford Nano Shared Facilities, Stanford University, Stanford, CA 94305-4088, USA; 3National Metrology Institute of Japan (NMIJ), National Institute of Advanced Industrial Science and Technology (AIST), Tsukuba Central 3, 1-1-1 Umezono, Tsukuba 305-8563, Japan; 4CREST-JST and School of Materials and Chemical Technology, Tokyo Institute of Technology, Ookayama, Meguro-ku, Tokyo 152-8550, Japan; kamegaki.s.aa@m.titech.ac.jp (S.K.); morikawa.j.aa@m.titech.ac.jp (J.M.); 5WRH Program International Research Frontiers Initiative (IRFI), Tokyo Institute of Technology, Nagatsuta-cho, Midori-ku, Yokohama 226-8503, Japan; 6Optical Sciences Centre (OSC), ARC Training Centre in Surface Engineering for Advanced Materials (SEAM), School of Science, Swinburne University of Technology, Hawthorn, VIC 3122, Australia; haoranmu@swin.edu.au (H.M.); danielsmith@swin.edu.au (D.S.); pmaasoumi@swin.edu.au (P.M.); 102932516@student.swin.edu.au (R.F.); tkatkus@swin.edu.au (T.K.); soonhockng@swin.edu.au (S.H.N.); 7Mark Wainwright Analytical Centre, University of New South Wales, Sydney, NSW 2052, Australia; sean.blamires@unsw.edu.au; 8School of Biological, Earth and Environmental Science, University of New South Wales, Sydney, NSW 2052, Australia; 9School of Mechanical and Mechatronic Engineering, University of Technology, Sydney, NSW 2007, Australia; 10Department of Mechanical Systems Engineering, Aichi University of Technology, Gamagori 443-0047, Japan; kondo-toshiaki@aut.ac.jp; 11Department of Electrical and Computer Engineering, Graduate School of Engineering, Yokohama National University, 79-5 Tokiwadai, Hodogaya-ku, Yokohama 240-8501, Japan; nishijima@ynu.ac.jp; 12Institute of Advanced Sciences, Yokohama National University, 79-5 Tokiwadai, Hodogaya-ku, Yokohama 240-8501, Japan; 13Research Institute of Electronics, Shizuoka University, Johoku 3-5-1, Hamamatsu 432-8011, Japan; moraru.daniel@shizuoka.ac.jp; 14Marine Biological Laboratory, University of Chicago, Woods Hole, MA 02543, USA; mshribak@gmail.com; 15College of STEM, School of Science, RMIT University, Melbourne, VIC 3000, Australia; elena.ivanova@rmit.edu.au; 16Department of Applied Chemistry, Tokyo Metropolitan University, Hachioji, Tokyo 192-0397, Japan; masuda-hideki@tmu.ac.jp

**Keywords:** anisotropy, polarization analysis, Stokes parameters, polarimetry

## Abstract

Herein, we give an overview of several less explored structural and optical characterization techniques useful for biomaterials. New insights into the structure of natural fibers such as spider silk can be gained with minimal sample preparation. Electromagnetic radiation (EMR) over a broad range of wavelengths (from X-ray to THz) provides information of the structure of the material at correspondingly different length scales (nm-to-mm). When the sample features, such as the alignment of certain fibers, cannot be characterized optically, polarization analysis of the optical images can provide further information on feature alignment. The 3D complexity of biological samples necessitates that there be feature measurements and characterization over a large range of length scales. We discuss the issue of characterizing complex shapes by analysis of the link between the color and structure of spider scales and silk. For example, it is shown that the green-blue color of a spider scale is dominated by the chitin slab’s Fabry–Pérot-type reflectivity rather than the surface nanostructure. The use of a chromaticity plot simplifies complex spectra and enables quantification of the apparent colors. All the experimental data presented herein are used to support the discussion on the structure–color link in the characterization of materials.

## 1. Introduction

It is well known that color and structure are closely linked. This link is established via the feature size and pattern (organization and alignment) [1]. Structural color, or the structural basis of color responsible for the vibrant shades observed on various insect species, can be endowed by the interaction of light (scattering and reflectance) with specific nano- and microstructures. The particular micro-nano surface structures (surface topography) also impart biomaterials with additional functionality, e.g., stronger light reflection or absorption due to the layered surface architecture of the feature arrangement along the height (a conifer or Christmas tree arrangement). Randomness, well-defined gratings, films, and cavities are key to the formation of structure-defined colors in nature [1], in addition to pigment (chemical)-based coloration. The lateral and axial dimensions of the absorbers and reflectors are discussed in electromagnetic (EM) antenna nomenclature. The portion of absorbed, reflected, and transmitted parts of light energy obeys energy conservation A+R+T=100%. One of the most relevant examples found in nature is the structural coloration of arachnid species such as Theraphosidae (tarantulas) and Salticidae (jumping spiders) [2] Some spiders are vividly colored using a combination of chemical and structural coloration strategies, providing unique examples for the design of photonic devices. With approximately 47,500 described species of spider from ∼100 different taxonomic groups [3], the understanding of their origin and function of structural coloration is still relatively unexplored. Furthermore, the correlation between specific feature alignment and color is obvious for characterization techniques using light within the visible spectral range. However, characterizing physical structures using different wavelengths (short UV and long IR-to-THz or sub-mm waves) allows us to confer new spectral properties. For example, recently, Kariko et al. established the origin of the red, silver, and black color phenomena observed on the theridiid spider, *Phoroncidia rubroargentea*, from Madagascar using complementary optical, structural, and chemical analysis [4]. Correlative structural analysis of complex 3D samples, such as spiders, is integral to defining the compositional origins of color. Specifically, surface topography at the nano- and microscales can be investigated via surface-sensitive optical techniques such as scanning electron and atomic force microscopy. Compositional features are important for the structural color phenomena that lie beneath the surface (e.g., the red color of the Theridiidae spider was found to be facilitated by a combined effect of multiple structural layers, including the thick sclerotized exoskeleton, guanine crystal microplates, and the chambered pigment-containing microspheres) are often investigated using artifact introducing, preparation-intensive techniques such as serial block face scanning electron microscopy, transmission electron microscopy, and confocal scanning laser microscopy. However, simpler optical characterization techniques, such as X-ray computed tomography and THz imaging, may be incredibly useful for structural characterization of color complexity. Indeed, the penetration depth is directly related to the wavelength. Currently, longer (mm) wavelengths being used for telecommunication networks and their interference, diffraction, and multiple reflections with materials are defined according to the optical properties (complex refractive index n+ik) of materials at those wavelengths.

Here, we overview several characterization techniques for the optical and X-ray characterization of natural materials. Specifically, we focus on the optical characterization techniques of the spectral properties of peacock spider (*Maratus volans*) scales and silk (spider silk and silk from the silkworm). The anisotropic nature of the microstructure of silk fibers is notoriously difficult to study [5]. Often, transmission studies of fibers suffer from reproducibility issues given the sample preparation and presentation complexity. For example, cutting the fibers introduces microstructural changes [6,7,8,9,10]. The thickness of the fibers may result in absorbance peak distortion, loss of resolution, and low signal-to-noise ratio. The orientation of the fibers relative to the beam also influences the spectral data, ensuring that all materials must be analyzed in the same orientation for reproducibility. Nevertheless, the optical and structural characterization of these natural materials is important to inform the design of biomimetic polymers/materials that possess enhanced structural and optical properties. Indeed, the superior features of silk produced by spiders in comparison to silk produced by silkworms may be informed by their specific nano- and microstructures, although both silks are composed of similar glycine-rich proteins. Due to the complexity of studying natural samples (shape and size complexity), the ability to achieve optical and structural characterization over several orders of magnitude of the features of interest is important from multiple aspects: (1) resolution and (2) composition/structure. The complexity of 3D surfaces and structures should be described in its entirety for an in-depth/comprehensive understanding of color and structure since the geometry and dimensions of 3D structures are of paramount importance for their color/spectral response and linked to mechanical and thermal properties [11].

## 2. Materials and Methods

The peacock spider specimens were kindly donated by Jürgen C. Otto. Spider preparation for scanning electron microscopy (SEM) observation involved the removal of the abdomen from the main body using a surgical-grade stainless steel scalpel. The abdomen was then immobilized on an Al stub using double-sided carbon tape. The spider was then fixed in 2.5% glutaraldehyde overnight at 4 °C, followed by washing with double-distilled H_2_O (3×5 min) and further fixation with 1% aqueous osmium tetroxide for 2 h. Routine ethanol (EtOH) dehydration was followed using 20%, 50%, 70%, 80%, 90%, 95%, and 100% ×2 EtOH. The abdomen was stored in EtOH prior to experiments. For SEM imaging, the fixed sample was placed in a critical point dryer (CPD; Polaron, E3100, Quorum Technologies Ltd.). The ethanol was gradually replaced with liquid CO_2_, which was then placed under supercritical conditions (1100 psi and 36 °C). Once dried, the sample was then coated with approximately 10 nm of gold prior to SEM imaging. SEM imaging was performed under high vacuum using the Raith direct-write electron beam lithography (EBL) system with high resolution (HR) field-emission imaging capabilities. Images of the abdomen were acquired at an accelerating voltage of 10 kV (see detailed discussion below).

Garden spiders (*Eriophora transmarina*) and golden orb web spiders (*Trichonephila plumipes*) were collected during the night from locations in Sydney, Australia. Their major ampullate (dragline) silk was collected by forcible spooling, as described by Blamires et al. [12]. The silks of this spider appear white to the naked eye, a consequence of high reflectance across the visible waveband outside of the UV [13]. Spiders were anesthetized using CO_2_, placed ventral side up on a foam platform, and immobilized using non-adhesive tape and pins. A single silk thread from the spinnerets was collected under a dissecting microscope. An electronic spool rotating at 1 m/min was used to reel silk threads from spiders [14].

Sericin cladding-free silkworm fibers used for optical imaging were prepared by established washing protocol in basic solution [15].

Focused ion-beam (FIB) milling of the abdominal platelets was applied to characterize the internal structure (IonLine, Raith). Rectangular sections were milled using a gallium (Ga) ion beam at 30 kV, 0.5 nA to a depth of sub-1 μm. The FIB cross-sections were then imaged by SEM for detailed observation of the internal structure of the platelets/scales.

### 2.1. Optical Imaging Techniques

Optical microscopy with polarization analysis was used for the optical characterization of the nanostructured surfaces of the spider scales and for silk samples. Emphasis was placed on imaging of the orientation azimuth [16]. A 4-polarization (4.pol) camera (CS505MUP1, Kiralux Polarization-Sensitive CMOS) was used for the detection of anisotropy in absorbance and retardance (Figure 1). For mapping the slow axCS505MUP1, iKiralux Polarization-Sensitive CMOSs orientation, we employed a recently invented polychromatic polarizing module (PPM; US patent 9625369). The PPM consists of a source of white light, a polychromatic polarization state generator, and a circular analyzer. While a traditional polarizing microscope generates the Newton interference colors if the specimen retardance lies from 300 nm to 2100 nm, the PPM creates the interference colors if the specimen retardance is from a few tenths of nm to 300 nm. The hue in PPM mostly depends on the slow-axis orientation rather than the retardance. This is in contrast to the traditional microscope, where the hue is determined by the retardance, but not the slow-axis orientation.

For reflection analysis of spider silk using a 4-pol. camera, an Au (60 nm thin film) mirror on a 5 nm Cr adhesion layer was thermally evaporated on cover glass at 60 A and 20 A currents, respectively, using LA-V5050L, Labotec evaporator at ∼4 × 10${}^{-4}$ Torr. We also used plastic λ/4-waveplates (Edmund Optics) as sample holders for transmission and reflection modes using the 4-pol. camera (Thorlabs).

### 2.2. X-ray Tomography

Hard X-ray micro-computed tomography (μ-XCT) was used for virtual sectioning of the reconstructed 3D volume of spiders. The μ-XCT experiment was carried out using ZEISS/Xradia Versa 520 X-ray microscope at the Stanford Nano-Shared Facilities, Stanford, CA. Both low- and high–low-resolution X-ray tomography volumes of the spider were collected with voxel sizes of 1.23 μm and 0.38 μm, respectively. The 3200 sample projection images were acquired in absorption mode using geometrical together with 4${}^{\times }$ and 20${}^{\times }$ optical magnification for low- and high-resolution modes, respectively. In order to remove the high-energy part of the X-ray spectrum, the X-ray source voltage was set to 30 kV, the minimum allowable for this instrument. To achieve the optimal signal-to-noise level (intensities of >5000 gray value over low transmission regions), the exposure time of 25 s was chosen. The μ-XCT datasets were reconstructed using the proprietary ZEISS Reconstructor software, Scout and Scan^TM^ Control System Reconstructor—16.1.13038.43540.

## 3. Results

### 3.1. Spider Scales: Peacock Spider

The male Australian Peacock (jumping) spider *Maratus volans* is ∼1 mm in size and shows very strong coloration (Figure 2). The abdomen of the tiny arachnid displays a striking pattern of red, blue, and black. The abdomen possesses microscopic scales that contain three-dimensional reflective diffraction grating structures. Some male peacock spiders are able to change their scales from red to green to violet with slight movements. In the family of structural color, the Morpho blue butterfly is the most well-known example of color being independent of the observation angle [17]. Structural color comes from the interference of light scattered from a surface array of nanostructures. Ordered nanostructures (such as grating patterns) produce small angle color changes (iridescence), whereas disordered nanostructures produce structural colors independent of the observation angle, similar to pigmented materials [18]. For the Morpho blue butterfly, its wide-angle color is attributed to a mixture of both ordered and disordered nanostructures in a hierarchical (multilayered) display [19].

The peacock spider has a distinct red pattern; its color is defined by the absorption of xanthommatin [20] contained within the brush-like structures, hence the coloration of a chemical origin (Figure 3a). The black color observed in some species is due to a combination of chemical and structural coloration [21]. For *M. volans*, the green-blue scales are composed of a colorless and transparent chitin with a refractive index of n≈1.5 [20] (Figure 3b).

Numerical simulations using the effective medium approach showed that the surface nanograting of ∼50 nm modulation depth and period ∼200 nm (Figure 3c,d) is mainly contributing as an anti-reflection structure, while the blue color is defined by the thickness of the chitin membrane (∼360 nm) and the gap (∼160 nm) between the two sides of the scale and the fiber structure of diameter 100 nm (with a surface filling ratio of 0.25) inside the air gap [20]. Effective medium theories (EMTs) relate the optical properties of the material with their composition and mass density and are very well suited for a specific, e.g., visible spectral range. EMTs can provide an analytical prediction of optical properties and are especially valuable due to simple estimates, which can be rigorously tested with numerically intensive finite-difference time-domain (FDTD) calculations. The effective refractive index can be calculated from the refractive indices and volume fractions of the given materials. Figure 4 shows the green-blue scale membrane milled by Ga ions to different depths. It reveals the fiber mesh on the inside section. In a separate experiment, the form-birefringent nature of the surface nanograting was revealed by the angular dependence of reflectance under illumination by linearly polarized light [20]. The effective medium theory of refractive index used in the modeling did not predict the polarization dependence [20]. There was no polarization dependence for the red brush structure, whereby the color originated from a chemical pigment (xanthommatin).

Figure 5 shows a 3D reconstruction of the micro-CT scan of the surface of the spider abdomen and respective cross sections in the XZ and YZ axis. The cross-sections do not reveal structures below ∼2μm; however, micro-CT is able to show the actual pattern of the scales on the abdominal surface. The scales cover the surface. In Figure 6, where the cross-sectional X-ray image is paired with the optical reflection image, the edges of the scales are well defined, while the middle parts have lesser brightness, and in the case of the optical image, clear interference patterns are recognizable. The central part of some of the scales is flatter (see Figure 4a), and the air gap inside the scale is smaller or absent. The high vacuum environment in the SEM and FIB chambers may have contributed to the flattening of the scales (Figure 2a).

The origin of the blue-green color appearance of the spider scales can be explained by the Fabry–Pérot (FP) etalon and its spectral selectivity. An FP plate of thickness *d* at a tilt angle β defines the effective thickness of the FP etalon, which becomes larger with tilt d/cosβ; the effective thickness and retardance are increased at an angle. The tilt angle tunes the phase delay δ=2πnd/λ. The transmission and reflection coefficients of the FP etalon are dependent on the reflectivity *R* of the FP film (chitin in the case scales) interfaces; both interfaces are assumed to have the same reflectivity. The FP transmittance and reflectance spectra are given by:(1)$$T_{FP}=\frac{{\left(1-R\right)}^{2}}{{\left(1-R\right)}^{2}+4R\sin^{2}\delta },$$
and complimentary reflectance (without absorbance A=0):(2)$$R_{FP}=\frac{4R\sin^{2}\delta }{{\left(1-R\right)}^{2}+4R\sin^{2}\delta },$$
respectively (widely used from visible to far-IR wavelengths [22]). The FP etalon imparts its reflectance/transmittance spectrum. The FP reflectance is maximum for a given *R* (of the single interface), when the phase delay corresponds to the λ/4 or δ≈π/2.

Next, the spider scale was modeled as two FP etalons separated by a sub-wavelength gap. The model was used to investigate the effect of color reflectance (Figure 3b). The scale itself is oriented at an angle to the exoskeleton, which can also contribute as a back-reflector (hence the transmitted spectrum is partially back-reflected). The twin FP structure of the scale and air gap is apparent from SEM images but is even better revealed using X-ray 3D tomography. Finite-difference time-domain (FDTD) calculations were implemented to explore predictions of the spider scale model.

Figure 7 shows the results of reflectance R(λ) and transmittance T(λ) spectra for the two FP chitin slabs separated with an air gap=160 nm and 50 nm (simulating a flattened scale), which are a closely matching model used in more elaborated calculations based on the actual structure of the spider scale and accounting for the effective refractive index on nanocorrugated surfaces [20]. The refractive index of chitin over the visible spectral range was calculated by Cauchy’s equation $n_{c}=A+B/\lambda^{2}$, where A=1.517 and $B=8.8\times 10^{3}$ [nm${}^{2}$] [20]. Calculations were carried out by the finite-difference time-domain (FDTD) method using the Lumerical software package (Ansys), 2022 R1 Finite Difference IDE. The total field scattered field monitors were set up to calculate the spectra; the incident field was linearly polarized, and the E-field strength was E=1 (Figure 7a). At the normal incidence, there is no difference for the s- and p-pol., which would change at a tilted angle (see Appendix C). The s-/p-pol. reflected and transmitted spectra become more complex, there are spectral regions where interference is causing intensity increase. However, all those complex spectra can be mapped onto a 2D chromaticity diagram, as shown next for the simpler case of normal incidence.

The R(λ) plotted on the chromaticity coordinates (x,y) shows color change upon the thickness of chitin increasing from 300 to 360 nm, which is consistent with the SEM and FIB analysis discussed above (Figure 7c). The two slabs of chitin define the color without sub-wavelength patterns on the outer and inner chitin walls. The polarization effect in R(λ) observed experimentally [20] is consistent with the pattern of nanogratings on the surface and the reflected light has a larger component of polarization normal to the scattering plane as follows from the Fresnel rules for the s- and p-polarizations. Spectra of R(λ) and T(λ) are complementary in terms of energy conservation (Figure 7b). The transmitted portion of light is larger than the reflected, understandably due to the low refractive index of chitin (close to that of window glass). This implies that the back-reflected light after transmission of a spider scale can produce the color appearance shown in (c). It is also noteworthy that the *R* and *T* points are located on the chromaticity (x,y) coordinates in such a way that they are on the opposite sides of the “white” central point (the highest color temperature), as shown in the inset of (c). This is another representation of the complementarity of *R* and *T* via the energy conservation A+R+T=100% in the absence of absorbance A=0.

### 3.2. Spider Silk

Spider silk, an excellent example of naturally occurring birefringence and alignment [23] was analyzed by polarization imaging techniques. Silk has a high ∼80% crystallinity [10] and is much less common in the water-soluble amorphous state [8], which is only attainable via a very fast thermal quenching of the disordered molten phase. Investigation of spider silks’ light absorbance and scattering over a broad UV–visible spectral range showed reduced absorbance at UV wavelengths [13]. Thus, the strong correlation between the glass transition temperature and the thermal degradation of silk with its color was experimentally established [14]. Figure 8 displays several possible optical imaging methods, including the typical transmission mode for intensity to cross-polarized imaging, which can be rendered into a color map of retardance, as well as a mapping of the orientation azimuth [16]. Silk fibers with structural defects that change the alignment of fibroin along the fiber direction can be observed using transmission mode imaging (Figure 9a); however, more detailed structural changes can be revealed using cross-polarized imaging showing localized changes of birefringence and retardance patterns. The structural color of the spider *Trichonephila plumipes* silk was further analyzed by the addition of a polychromatic polarization module to a standard benchtop optical microscope. The PPM add-on allows visualization of the birefringence instantly and independently of the specimen orientation. In Figure 8, the hue represents the orientation of the slow axis, and the saturation depicts the retardance amount. The dependence of the hue on the orientation of the slow-axis φ can be approximated by a linear function $Hue=2(\varphi +\varphi_{0})$, where $0^{\circ }\leq Hue<360^{\circ }$ with $0^{\circ }\leq \varphi <180^{\circ }$ and $\varphi_{0}$ is a constant defined by a reference angle, which depends on the mutual orientation of the polarization state generator and the camera. In order to find $\varphi_{0}$, we can use a birefringent structure with retardance of approximately 30 nm and the slow axis oriented at $0^{\circ }$. It is convenient to employ a calibration test target, which was developed by Prof. Peter G. Kazansky (University of Southampton). The test target has a birefringent star pattern with slow axes oriented in the radial direction. The birefringent star is shown in the top right corner of the polychromatic polarization image in Figure 8. The reference angle $\varphi_{0}$ equals half of the hue of the horizontal wedge. Then we can compute a map of the slow axis distribution as $\varphi =Hue/2-\varphi_{0}$. It is necessary to mention that the linear approximation can introduce an error of measured slow-axis orientation, about $3^{\circ }$. In order to suppress the error, we can build a calibration curve by measuring the hue of each wedge. Another option is to mechanically rotate the birefringent specimen and measure the hue at each azimuthal position. Thus, the color-azimuth map is an additional useful presentation of structural changes, as discussed in this article. Figure 10 shows optical images in different presentations for a thicker *Antheraea pernyi* moth silk.

Next, experimental determination of the absorbance (Equation (Equation 4)) of spider (*Trichonephila plumipes*) silk (yellow) using a 4-pol. camera at the OUT port (Figure 11) and the transmittance T(θ) fit by Equation (Equation 5) (without retardance contribution) was performed with the same setup and the same number of optical elements in the beam, only changing their azimuthal orientation (Figure 11). For the non-polarized incident illumination, the spider silk was placed on an Au mirror (Figure 1a). Three polarizer positions on the 4-pol. camera were used for the fit of reflected light using $R\left(\theta \right)=Amp\times \cos\left(2\theta -2\theta_{shift}\right)+Offset$; normalization of the reflected signal with silk was carried out using the reflection from the Au mirror for the reflectance R(θ) at four individual orientations (Figure 1b). The 4-pol. camera image at crossed Nicol position ($0^{\circ }$-segment) was not used for the fit due to low intensity. It is noteworthy to add that simultaneous fit by Equation (Equation 5) would require at least six independent data points (six polarization orientations). This is impossible without rotation of the sample or polarizers since only four orientations are measured instantaneously with a 4-pol. camera. Such rotations cause an image shift and strongly compromise the fidelity of the fit [25]. Figure 1b shows maps of the three best-fit parameters. The phase map $2\theta_{shift}$ shows that the main part of the fiber has the $0^{\circ }$-azimuth along the length direction as expected. Cross-polarized imaging was also carried out (not shown) since such a setup is sensitive to retardance anisotropy and birefringence. However, the fit was inconclusive, most probably due to the larger angular frequency of the signal, which was poorly resolved by the three-point fit.

## 4. Discussion

Optical anisotropies related to the real and imaginary parts of the refractive index (n+ik) are linked to the birefringence Δn and dichroism Δk. Both Δn and Δk occur for circular and linearly polarized light, depending on the material order and structure. The simplest measurement of absorption anisotropy can be made by using one linear polarizer, which is used to control the polarization of the incident light. For the birefringence, two polarizers (a pair of polarizers and an analyzer) must be used. In such cases, the transmitted intensity follows θ-azimuthal dependence when absorbance and reflectance are negligible (R≈0, A≈0):(3)$$T\left(\theta \right)=\sin^{2}\left[2\left(\theta -\theta_{ret}\right)\right]\sin^{2}\left(\pi \Delta nd/\lambda \right),$$
where θ is the orientation angle, $\theta_{ret}$ is the slow or fast axis direction (i.e., the slow axis is usually aligned to the main molecular chain or along a polymer stretch direction), and Δn is the birefringence of the sample/object at the wavelength λ for the thickness *d*. This equation defines the Maltese cross with the dark intensity positions at $\theta =\theta_{ret}$ and $\theta =\theta_{ret}\pm \pi /2$, while the most bright regions are at $\theta =\theta_{ret}\pm \pi /4$. The phase retardance δ=2πΔnd/λ is defined by the second sin term in Equation (Equation 3).

### 4.1. Polychromatic Polarizing Module

The polychromatic polarizing module (PPM) is an add-on that can be used with a regular benchtop optical microscope. The main component of the PPM is a polychromatic polarization state generator, which produces polarized light with the polarization ellipse orientation determined by the wavelength. A set of ellipses corresponding to different wavelengths is called a spectral polarization fan. All polarization ellipses have the same ellipticity angle. If there is no alteration of the beam polarization by a specimen, then all wavelengths are transmitted evenly to the circular analyzer. As a result, we see a gray background. If the major axis of the polarization ellipse is at $45^{\circ }$ or $135^{\circ }$ to the slow axis of a birefringent specimen, the intensity of light transmitted by the circular polarizer will be minimal or maximal, respectively. For example, if the major axes of red and green polarization ellipses are oriented at $135^{\circ }$ and $45^{\circ }$, respectively, and the specimen slow axis is oriented at $0^{\circ }$, then transmission of the red wavelength will be maximal, and transmission of the green wavelength will be minimal. As a result, the specimen will be red. In the case of rotation of the specimen by $90^{\circ }$, the situation will be reversed, and the specimen will be green.

### 4.2. Four-Polarization Camera

Biomaterials, such as silk, can exhibit not only linear but also circular dichroism due to specific protein structures. Measuring optical anisotropy using linear as well as circularly polarized light can be carried out with a simple setup assembled on the microscope, as shown in Figure 11. The setting of the circular right-/left-pol. is performed by the orientation of the linear polarizer at the IN port. If the transmission axis (orientation of the E-field) of the linear polarizer is aligned with the slow axis of the λ/4 waveplate, a linearly polarized light is launched from the IN port. Different optical characterizations can be possible depending on the polarization elements at the detection OUT port, as discussed next. In all cases, we consider the 4-pol. camera as a detector, which already provides four orientation analyses of the sample (reflection and absorption for transparent samples, Figure 11).

By capturing an image with a camera (e.g., CS505MUP1 Thorlabs) with four directional wire-grid arrays integrated into the CMOS sensor, polarization analysis can be made faster. Even more importantly, the image shifts and distortions when the polarizer (or sample) are rotated are significantly reduced to acquire several images to fit transmittance *T* (or absorbance *A*) by a harmonic function [25]. For the absorber oriented at angle $\theta_{abs}$, the absorbance $A=-\log_{10}T$ is defined [27]:(4)$$A\left(\theta \right)=\frac{A_{max}-A_{min}}{2}\cos\left(2\theta -2\theta_{abs}\right)+\frac{A_{max}+A_{min}}{2},$$
where measurement is carried out with incident light at four selected polarizations, and detection is not discriminated in polarization. With a 4-pol. camera, four images are directly acquired in a single acquisition while the incident light is non-polarized (isotropic random). The setup is shown in Figure 11 without λ/4 waveplates at the IN and OUT ports; polarization homogenizers used at the IN port of some microscopes can be useful for polarization-isotropic illumination and are made with circular polarizers (λ/4 plates based on optical activity; not shown in Figure 11). The fit can also be modeled by the sin function with freely chosen phase sign $\pm \theta_{abs}$, without loss of generality (±cosθ=sin[θ±π/2]). Such measurement reveals the anisotropy of absorption for linearly polarized light (it can also be measured in the transmission rather than reflection mode, as shown in Figure 11). Since polarization is only set at one of the two IN/OUT ports, such measurements are not sensitive to polarization changes due to retardance Δn×d, where *d* is the thickness of the sample.

Setting circular polarization at the IN port (RHC or LHC) and without a λ/4 waveplate before the 4-pol. camera makes it possible to measure the absorbance *A* of an image using Equation (Equation 4).

With linear polarization at the IN port (λ/4 and linear polarizers are both aligned at $0^{\circ }$) and no λ/4 plate at the OUT port, a setup to measure birefringence and absorbance anisotropies is realized [28], i.e., a typical polarizer-analyzer arrangement (Equation (Equation 3)) only using 4-pol. camera. Since absorbance is π-fold (equal absorbance at 0 and π), while birefringence has a twice-high angular dependence, the fit function to account for the two contributions in transmittance is conveniently chosen:(5)$$T\left(\theta \right)=\left[a_{\kappa }\cos^{2}\left(\theta -b_{\kappa }\right)+o_{\kappa }\right]+\left[a_{n}\cos^{2}2\left(\theta -b_{n}\right)+o_{n}\right]\equiv Abs+Ret,$$
where $a_{\kappa }$ and $a_{n}$ are the amplitudes related to absorbance Abs and retardance Ret contributions, $b_{\kappa }$ and $b_{n}$ are the orientation-dependent angles (which can be different for the two anisotropies), and $o_{\kappa }$ and $o_{n}$ are their corresponding offsets. The first term $\left[a_{\kappa }\cos^{2}\left(\theta -b_{\kappa }\right)+o_{\kappa }\right]$ is equivalent to Equation (Equation 4) by use of the identity $\cos2\theta =2\cos^{2}\theta -1$, and both define the anisotropy of absorbance (the π-folding in angular dependence). Three separate measurements are required to fit the function with the three fit parameters, i.e., three polarization angles from the 4-pol. camera image are required and are sufficient.

The second $\left[a_{n}\cos^{2}2\left(\theta -b_{n}\right)+o_{n}\right]$ term is due to retardance (birefringence) and has a twice-larger angular frequency, i.e., π/2-folding (Equation (Equation 3)). Additionally, three separate angles of polarization are enough for the fit function; however, such a fit usually returns a lower confidence range due to the larger angular frequency for retardance. In many practical cases, one of the two Abs or Ret parts dominates the measured transmittance T(θ). For example, at the infrared (IR) molecular fingerprinting spectral range, absorption bands tend to dominate and the retardance effects are small [29]. It is also clear from Equation (Equation 5) that by fitting one of the two θ (Abs) or 2θ (Ret) dependencies, the other is ignored, while the measured one is usually overestimated.

### 4.3. Stokes Parameters from 4-pol. Imaging

Four Stokes parameters define the state of polarization and aim to fully characterize the detected light (coherent and incoherent). Firstly, three Stokes parameters per pixel can be calculated from the measured intensity 4-pol. images using simple image algebra. The intensity or $S_{0}=\left(I_{0}+I_{\pi /4}+I_{\pi /2}+I_{3\pi /4}\right)/2$, $S_{1}=I_{0}-I_{\pi /2}$, $S_{2}=I_{\pi /4}-I_{3\pi /4}$. The best fit of 4-pol. transmission measurements is usually fitted by generic expression $Amp\times \cos(2\theta -2\theta_{shift})+Offset$. The azimuth then is $\theta_{shift}=\arctan_{2}\left(S_{2},S_{1}\right)/2$, where $\arctan_{2}$ is the four quadrants’ inverse tangent. Additionally, the degree of linear polarization can be calculated as $DoLP=\sqrt{S_{1}^{2}+S_{2}^{2}}/S_{0}$. The DoLP is widely used for edge detection in machine vision.

The intensity of transmitted light through a λ/4-waveplate at angle ϕ and a polarizer/analyzer at θ is given by [30]:(6)$$I_{T}\left(\theta ,\phi \right)=\left[S_{0}+S_{1}\cos\left(2\theta \right)+S_{2}\sin\left(2\theta \right)\cos\phi -S_{3}\sin\left(2\theta \right)\sin\phi \right]/2.$$

From four independent measurements all Stokes parameters are obtained [30]:(7)$$\begin{array}{cc} S_{0} & =I(0,0)+I(\pi /2,0), \end{array}$$(8)$$\begin{array}{cc} S_{1} & =I(0,0)-I(\pi /2,0), \end{array}$$(9)$$\begin{array}{cc} S_{2} & =2I\left(\pi /4,0\right)-S_{0}, \end{array}$$(10)$$\begin{array}{cc} S_{3} & =S_{0}-2I\left(\pi /4,\pi /2\right). \end{array}$$

For determination of $S_{3}$, circularly polarized light is required and can be generated by adding a λ/4 waveplate with a slow/fast axis at π/4 degrees to the incident linear polarized light $S_{3}=I_{\pi /4}^{\lambda /4}-I_{-\pi /4}^{\lambda /4}$. Yet, a simpler method to obtain all four Stokes parameters $S_{0,1,2,3}$ is useful with four independent measurements. First, the polarizer is set at θ=0,π/4,π/2, and then a λ/4-waveplate is added at the π/4-orientation for the fourth measurement of intensity. The fourth Stokes component can be calculated as $S_{3}=S_{2}\times \tan\delta$, where δ is the phase retardance $\delta \left(\lambda \right)=\frac{2\pi d}{\lambda }\left[n_{e}-n_{o}\right]$ for thickness *d*. The first three are directly measured from 4-pol. images using simple image algebra. The intensity or $S_{0}=\left(I_{0}+I_{\pi /4}+I_{\pi /2}+I_{3\pi /4}\right)/2$, $S_{1}=I_{0}-I_{\pi /2}$, $S_{2}=I_{\pi /4}-I_{3\pi /4}$.

### 4.4. Nanotextured Surfaces for Analysis of Polarization Anisotropy in Reflection

Time-dependent evolution of changes in optical anisotropies using a 4–pol. camera is advantageous since all four images at each separate 45${}^{\circ }$ degree change in polarization are obtained in one acquisition. Such anisotropic azimuth can be calculated from Stokes parameters $S_{1}$ and $S_{2}$ as well as DoLP, which shows the edge and is widely used in machine vision. Moreover, the polarization analysis can resolve the orientation and alignment of features within spectral regions that are up to $20^{\times }$ smaller than the diffraction limit, as was demonstrated for the IR spectral range using microscopy [31]. Additionally, the same principle of discerning orientation when spatial resolution is beyond the required feature size was demonstrated with a 4-pol. camera attached to a drone from a 20 to 140 m height [32]. Figure A2d shows another example where the sub-wavelength (nanoscale) feature of a diatom is not resolved and is beyond the diffraction limit, e.g., ∼0.5 μm; however, using a 4.pol camera, the nanofeatures are resolved by color. The nanoslots (rectangular voids ∼5 μm in length, ∼1 μm in diameter) show strong polarization anisotropy in transmission [33]. The azimuth of the colored regions in diatoms was only 13${}^{\circ }$ for each separate and distinct color. An image of a nanostructured surface in reflection mode (Figure A4a) can be analyzed for optical anisotropies by the method outlined above. Shape changes of the observed object contribute to optical changes (anisotropy) that can be determined by the 4-pol. method. Thus, real-time monitoring of cell attachment, growth, or mechanical changes to the cell membranes can potentially be monitored by a 4-pol. camera. This is an important in situ characterization technique currently unexplored for assessment of the mechano-responsiveness of cells to nanomaterials.

The conjecture we present here is that 4-pol. imaging can be used to trace changes to the shape of various cell types attached onto nanotextured surfaces and to correlate those changes to their mechano-responsiveness. Additionally, 4-pol. imaging of the mechano-transduction of cells in response to nanotextured surfaces can be supported by the anti-reflection property of such surfaces [34]. A gradual change in refractive index at the liquid–material interface decreases light reflection, which is valuable for achieving better contrast in imaging. For example, nanotextured silicon (Si; black-Si [35]) alters the gradient forces acting on polymeric and gel chains within focal spots of 1 micron in diameter [36] and could potentially be used for color mapping using a chromaticity plot, as introduced earlier.

## 5. Conclusions and Outlook

In this article, we present an analysis of experimental results showing intuitive color visualization of the 3D complex structures of natural materials such as spider scales and silk retrieved from polarization analysis using optical microscopy. Chromaticity coordinates (x,y) are useful to trace (and quantify) minute color changes. We show that the simple sub-wavelength grating on twin slabs of chitin, which compose the spider cuticle, can produce vivid colors as demonstrated using a simple model linking reflectance and transmittance spectra. Visualization using a chromaticity diagram shows the trends of color changes according to geometrical structure parameters in the (x,y) chromaticity presentation. Additionally, polarization effects due to s- and p- polarizations can be more intuitively understood using a chromaticity plot, as compared with the spectral presentation of *T* or *R*.

We present an optical characterization method by which the differences in the orientation of anisotropic structures are proportional to the perceived differences in color. It is shown that a 4-pol. camera allows the acquisition of simultaneous images for calculating three Stokes parameters $S_{0,1,2}$ and the azimuth of retardance, which can also be directly imaged by a polychromatic polarization method using a standard optical microscope. The 4-pol. method has the inherent capability to determine structure orientation and its anisotropy below the spatial resolution [37]. It was shown that this technique is applicable not only to the optical far-fields [28] but to the non-propagating near-fields widely adopted for attenuated total reflection (ATR) spectroscopy. Imaging of the orientation of absorbers and retardance below the surface of the samples, including biological tissues, has broad application potential, especially at the long IR and THz spectral range [38].

Finally, the demonstrated polarization and color analysis of *R* and *T* spectra can be harnessed in the currently active area of research on “mechano-biocidal surfaces” [39,40,41,42]. Mechano-biocidal surfaces are so named because they have been demonstrated to mechanically rupture microbes that encounter the surfaces. The contact-killing surfaces usually exhibit an array of high-aspect-ratio nanofeatures (nanotopography). The polarization analysis of the nano-micro features on nanotextured surfaces is still relatively unexplored. However, (auto)fluorescence, and absorbance changes, which can be plotted onto chromaticity color maps, can reveal new structural information in addition to the standard topographic analysis currently performed for nano-microstructured surfaces.

Spider silks are among the most mechanically resilient fibers known to man. In the future, experimental stability testing of spider silks in outer space conditions (e.g., exposure to microgravity, cosmic radiation) will be performed, and the evaluation will be based on its high refractive index and waveguiding properties (Appendix E). We hypothesize that the degradation of hydrogen-bonded molecules can be monitored via color and intensity mapping using simple observation methods during the harsh outer space environment. 

## Figures and Tables

**Figure 1 nanomaterials-13-01894-f001:**
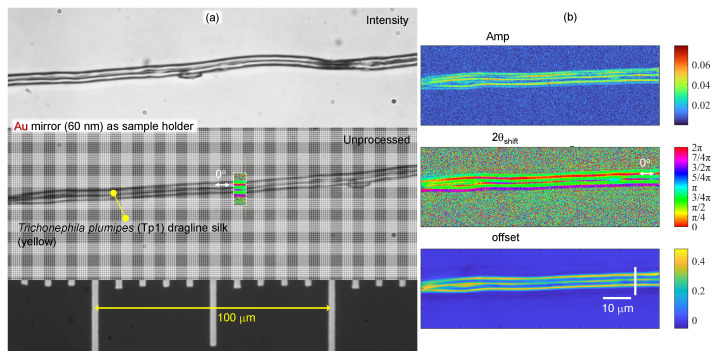
*Trichonephila plumipes* (Tp1) dragline silk. (**a**) Optical images captured by the 4-pol. camera as average (intensity) and unprocessed (intensity of separate pixels without averaging) using an objective lens with $40^{\times }$ magnification and NA=0.74 (Olympus). Illumination was non-polarized. Silk fibers were placed directly on the Au mirror (60 nm thickness was evaporated on cover glass with 5 nm Cr adhesion layer). The overlaid color segment on the fiber shows the corresponding phase $2\theta_{shift}$ structure. (**b**) Selected region was fitted by $Amp\times \cos(2\theta -2\theta_{shift})+Offset$. The fit parameters’ maps $Amp,2\theta_{shift},Offset$. See Figure A2c for structure of the spider silk.

**Figure 2 nanomaterials-13-01894-f002:**
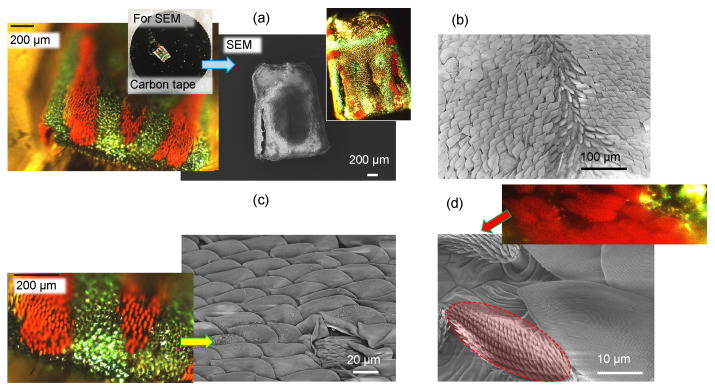
Peacock (jumping) spider. (**a**) Low-magnification optical and SEM images of the excised spider abdomen. Inset image shows the dissected spider abdomen fixed onto carbon tape for SEM imaging. The vibrant array of colors (blue-green and red) is produced by structural coloration and pigment, respectively. (**b**) Low-magnification SEM image of the scales that possess the structural green-blue color. The image was taken along the central long axis of the abdomen, showing the changing orientation of the scales toward the center of the spider abdomen. (**c**) High-magnification SEM image of the green-blue scales. (**d**) SEM image of the red brushes (pigment coloration). The Inset image shows a high-resolution optical image, revealing the vibrant red color.

**Figure 3 nanomaterials-13-01894-f003:**
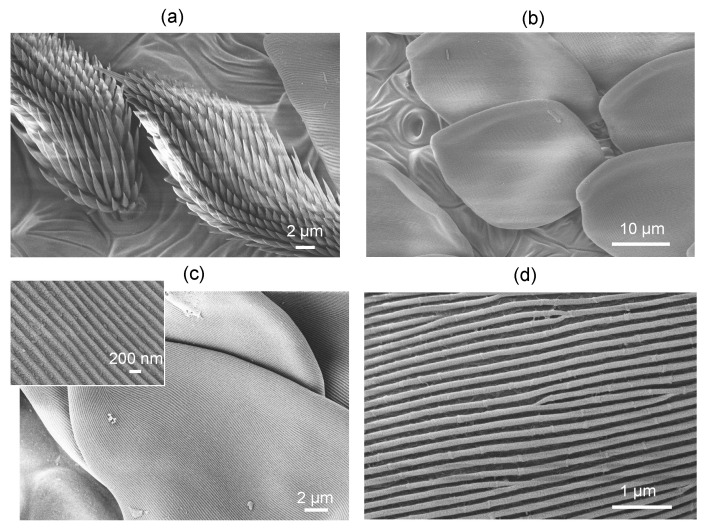
Peacock (jumping) spider. (**a**) SEM images of brushes with red coloration. (**b**) SEM image of the green-blue scales. (**c**,**d**) High-magnification (∼2 × 10${}^{4}$) SEM images of the individual scales and the surface grating structure that causes the blue-green structural color.

**Figure 4 nanomaterials-13-01894-f004:**
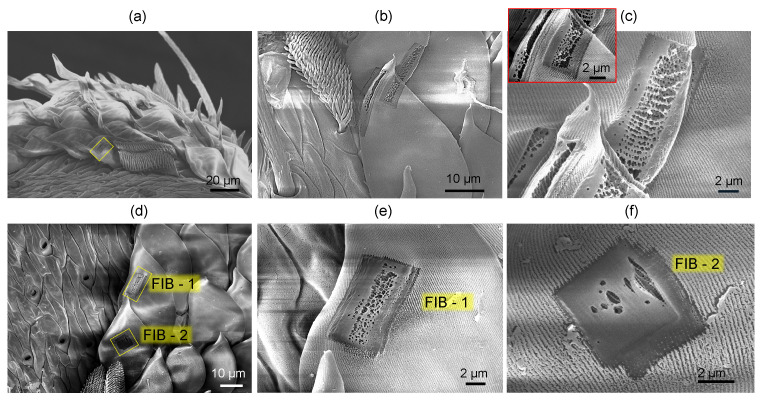
SEM images of FIB-milled green-blue scales at different magnifications: overview image (**a**), single scale (**b**), FIB milled region at the edge of the scale (**c**), two milled regions FIB-1 and FIB-2 at small (**d**) and large (**e**,**f**) magnifications. Chitin membrane of the green-blue scale has thickness ∼350 nm and an air gap inside.

**Figure 5 nanomaterials-13-01894-f005:**
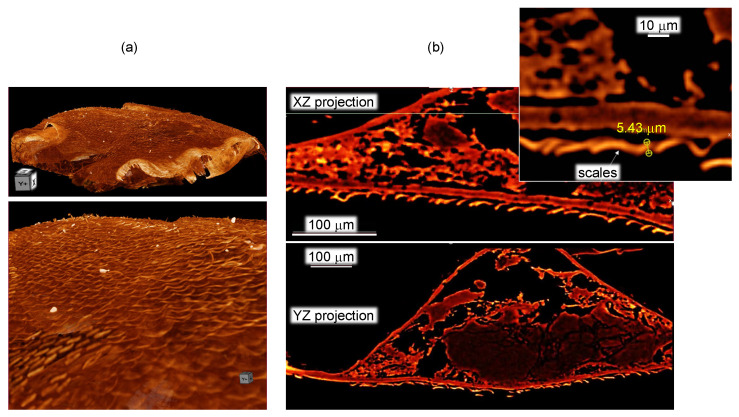
X-ray tomography of Peacock spider abdomen region. Large scale (**a**) and 3D tomography cross sections (**b**). Movies of the 3D sectioning XZ and YZ are added in Appendix A.

**Figure 6 nanomaterials-13-01894-f006:**
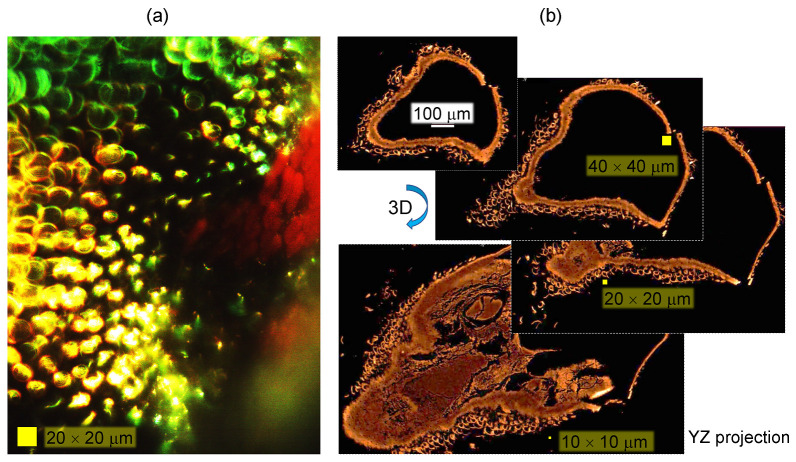
Optical reflection (**a**) and X-ray (**b**) resolved the green-blue scales of the Peacock spider (same specimen as in Figure 5 only projection is showing a flat cross-section of the scales). Movies of the 3D sectioning shown in (**b**) are added in Appendix A.

**Figure 7 nanomaterials-13-01894-f007:**
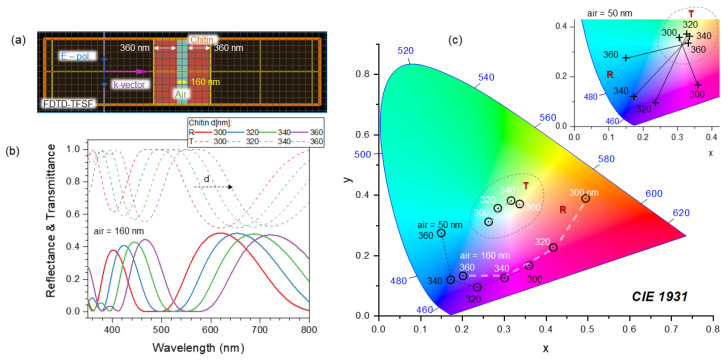
(**a**) Finite-difference time-domain (FDTD) 2D model of two Fabry–Pérot (FP) slabs of chitin of thickness *d* with nanogap of gap=160 nm. The total-field/scattered-field (TFSF) model was set up for the calculation of the reflectance *R* and transmittance *T* spectra shown in (**b**). The color appearance of the reflection from the chitin FP slabs separated by d=160 nm gap are shown by white *d*-labels and the line in (**c**). Absorbance A=0, which defines R+T=100%. (**c**) Chromaticity diagram for the color appearance of reflection spectra of two coupled FP slabs with an air gap of 50 (black lines) and 160 nm (white lines). Additionally, the transmittance *T* for d=160 nm gap is shown in the dashed-line encircled region. Inset shows *R*−*T* points connected by lines for the same *d* when the air gap was 50 nm (see Appendix B for different angles of incidence in Figure A3).

**Figure 8 nanomaterials-13-01894-f008:**
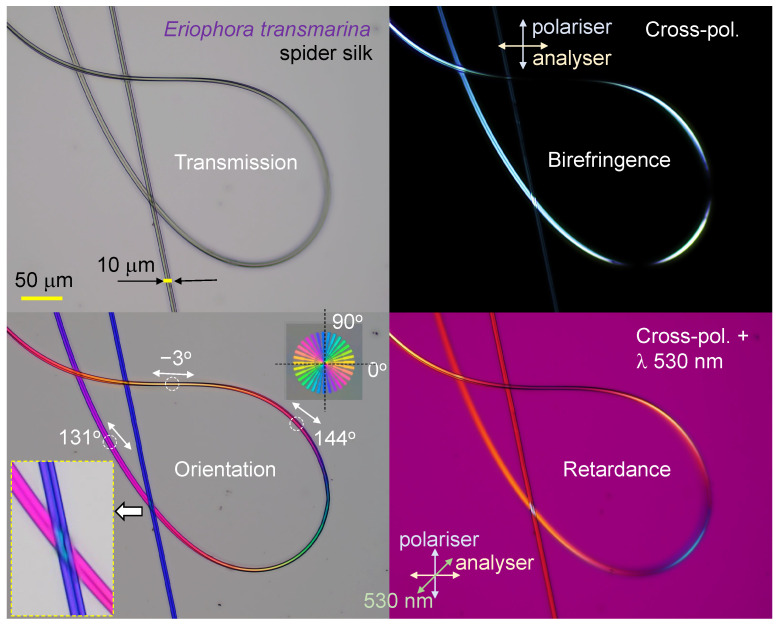
Optical microscopy images of spider (*Eriophora transmarina*) silk harvested at constant mechanical pulling force. Images were taken in transmission mode and show intensity, birefringence, retardance, and azimuth (of the slow axis). For the retardance image, the full-wavelength (530 nm)λ plate was at the orientation (looking at the image) from SW-NE. The slow-axis orientation is calculated as $Azimuth=\left[Hue/2\right]-20^{\circ }$, where $0^{\circ }<Hue<360^{\circ }$. The slow-axis color reference shows the orientation angles; the reference sample with an fs-laser-inscribed nanograting inside silica without surface damage [24] was made by Prof. P. G. Kazansky’s team.

**Figure 9 nanomaterials-13-01894-f009:**
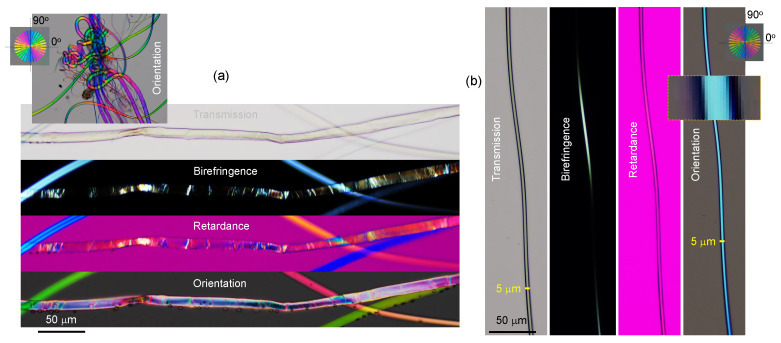
Optical microscopy images in transmission mode showing intensity, birefringence, retardance, and azimuth (of the slow axis) at the complex (**a**) and homogeneous (**b**) sections of spider silk. Inset in (**a**) shows an entangled region of silk fiber and wishers shedding off the fiber (see Figure A1 for detailed optical analysis).

**Figure 10 nanomaterials-13-01894-f010:**
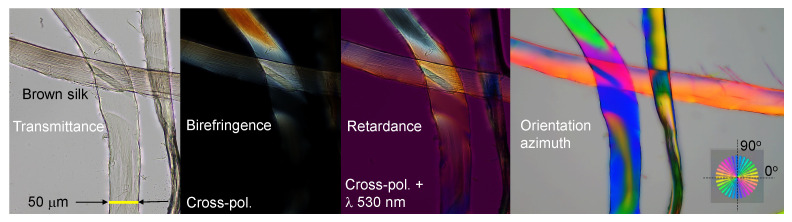
Optical characterization of *Antheraea pernyi* moth produced silk after degumming [15]. Single fibroin strands were imaged. Silk samples were kindly provided by Prof. Jingliang Li.

**Figure 11 nanomaterials-13-01894-f011:**
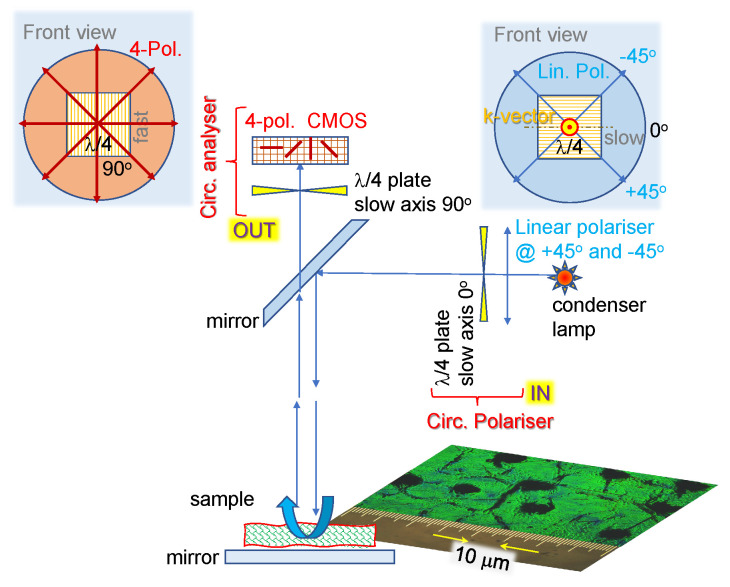
Universal setup for detection of absorption and reflection/scattering differences between the left- and right-hand circular polarizations (LHC/RHC), the optical activity and circular dichroism, using a 4-pol. camera (see text for discussion). The front-view insets show orientations by looking into the beam. The two λ/4 waveplates have slow-axis crossed and impart canceling contributions to the beam phase ±π/2 and ∓π/2, correspondingly to the position of the linear polarizer at the input with $\pm 45^{\circ }$ angle with the slow axis of the λ/4 plate at the first polarizer plate. The inset shown is a sample image from the wing of a Japanese jewelry bug (*Chrysochroa fulgidissima*, which has anisotropy for RHC and LHC polarizations [26]).

## Data Availability

Data produced in this study can be made available upon a reasonable request.

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
