# Peer review of "Structure and Optical Anisotropy of Spider Scales and Silk: The Use of Chromaticity and Azimuth Colors to Optically Characterize Complex Biological Structures"

_nanomaterials, 2023, doi:10.3390/nano13121894_

Round 1
Reviewer 1 Report
In this work the authors have set the ambitious goal to experimentally demonstrate the “structure-color link in the characterisation of materials”. Indeed, a complex characterization approach is applied, using various techniques (SEM, X-ray tomography, optical polarization imaging, etc.) and broad range of length scales – from nm to mm. Mainly spider scales and silk are investigated, but some results are also presented about metallic and polymer nano-pillars.
The study is interesting and diverse; however, I have some comments and questions that should be addressed by the authors:
- The title does not fully cover the range of results presented in the manuscript. Some revision is advisable.
- The model of the 4-polarisation camera should be given in Section II.A.
- The Results and Discussion sections are not clearly differentiated. There are many results presented in the Discussion section, as well as description of experimental methods, that would better fir either the Materials and Methods or Results section. I believe, certain restructuring of the text would make it much easier to follow and comprehend.
- I don’t clearly see why the nano-pillars are included in this work. How they relate with the spider scales and silk studies?
- Finally, because of the large number of authors, I would recommend to include an Author contributions section, where the contributions of each author can be outlined and distinguished.
Author Response
The study is interesting and diverse; however, I have some comments and questions that should be addressed by the authors:
- The title does not fully cover the range of results presented in the manuscript. Some revision is advisable.
We have revise the title to:
Structure and optical anisotropy of spider scales and silk: the use
of chromaticity and azimuth colors to optically characterise
complex biological structures
- The model of the 4-polarisation camera should be given in Section II.A.
Corrected, as suggested.
- The Results and Discussion sections are not clearly differentiated. There are many results presented in the Discussion section, as well as description of experimental methods, that would better fir either the Materials and Methods or Results section. I believe, certain restructuring of the text would make it much easier to follow and comprehend.
We thank the reviewer for their comment. We have moved the section describing the optical characterisation of silk using the PPM and 4. Pol imaging of the spider silk to the relevant part of the results section.
- I don’t clearly see why the nano-pillars are included in this work. How they relate with the spider scales and silk studies?
In the discussion section, we hypothesise that, using the 4 pol. Camera we can detect changes in optical anisotropies on nanostructured surfaces related to the change in shape of cells attached to such surfaces. This is an important in situ characterisation technique currently unexplored for assessment of the mechano-responsiveness of cells to nano-materials. For clarity and flow, we have removed the mat and methods section for the fabrication of the nanopillars to supplementary, along with figure 12.
- Finally, because of the large number of authors, I would recommend to include an Author contributions section, where the contributions of each author can be outlined and distinguished.
We have included an author contribution section.
Reviewer 2 Report
See the attached file.

Author Response
The paper by Denver Linklater, et al. is concerned with an overview of numerous poorly researched structural and optical characterization techniques applied to biomaterials. The various techniques are extensively discussed and thus provide new insights, which will be valuable for researchers working in this field. Of course, some questions/remarks remain, which are formulated below:
- On p.6, 2 question marks appear 3 lines from above. …violet with slight movements??. These question marks must be made specific. Are they typo’s?
It was a reference incorrectly cited in latex. now fixed.
- Just above th Fig.1 the authors state: The medium theory of refractive index.. Do the authors mean: "Effective medium theory"?
Corrected as suggested.
- The effective medium theory is used quite successfully for the description of many of the phenomena mentioned in this paper. A discussion of this powerful technique is missing and would certainly contribute to the understanding of all phenomena discussed in this paper. Once these questions/remarks stated above have been answered, the paper will give an important contribution to this interesting field of physics and is certainly suitable for publication in your journal.
As suggested, we have added a sentence explaining the EMT is added to the results section.
‘Effective medium theories (EMT) relate the optical properties of the material with
their composition, mass density and are very well suited for a specific, e.g., visible spectral range. The EMTs can provide analytical prediction of optical properties and are especially valuable due to simple estimates, which can be rigorously tested with numerically intensive finite difference time domain (FDTD) calculations. The effective refractive index can be calculated from the refractive indices and volume fractions of the given materials.’
Reviewer 3 Report
Reviewer’s comments on manuscript „Structure and optical anisotropy of spider scales and silk: (nanomaterials#2385101)” by D. Linklater et al.
This is a rather well written review about the state of the art of usage of a wade range of polariscopy techniques – ranging from Fabry-Perot reflectance imaging to Stokes-parameter imaging. The manuscript nicely illustrates the ways how superresolution can be obtained by combining the different polarization modalities. As an illustrating example the relationship between the optical properties and biological structure of spider scale and silk is discussed. Because the illustrated optical methods can be used in in-vivo imaging of biological samples and consequently may find applications in biomedical diagnoses of different illnesses, research in this field are not only interesting but also important.
I have found only a few subjectal and formal shortcomings, listed below.
Subjectal concerns:
1. On Page 15, please rewrite in the text jut above Eq. 6, the mathematical formula for thetashift. In the present form it is not understandible. The meaning of the unconventional „arctan2(S2, S1)/2” is not clear, without a detailed definition.
2. In the text, the place of citation for the reference paper #41 can not be found.
3. Legend of Fig. 2 tells about „high magnification SEM images”. Please append these magnification values.
Formal concerns:
1. On Fig. 7 separation of the whole image into the 4 subimages (labelled by a-b-c-d) is suggested, with the appropriate modification of the legend.
2. Please spell-check the text for typographical errors.
Because of the complexity of this review, the repetition of the verbal definitions of the main optical methods, separetely in a Glossary, is suggested.
A spell-checking is suggested, there are several typografical errors, mis-writings.
Author Response
This is a rather well written review about the state of the art of usage of a wade range of polariscopy techniques – ranging from Fabry-Perot reflectance imaging to Stokes-parameter imaging. The manuscript nicely illustrates the ways how superresolution can be obtained by combining the different polarization modalities. As an illustrating example the relationship between the optical properties and biological structure of spider scale and silk is discussed. Because the illustrated optical methods can be used in in-vivo imaging of biological samples and consequently may find applications in biomedical diagnoses of different illnesses, research in this field are not only interesting but also important. Answer.
Thank you for the positive evaluation.
I have found only a few subjectal and formal shortcomings, listed below.
Subjectal concerns: On Page 15, please rewrite in the text jut above Eq. 6, the mathematical formula for thetashift. In the present form it is not understandible. The meaning of the unconventional „arctan2(S2, S1)/2” is not clear, without a detailed definition.
Answer. Revised. The arctan_2 is standard four quadrants inverse tangent.
In the text, the place of citation for the reference paper #41 can not be found.
Answer. Revised. It is in conclusions.
Legend of Fig. 2 tells about „high magnification SEM images”. Please append these magnification values.
Answer. 20,000 times. Added in captions.
Formal concerns: On Fig. 7 separation of the whole image into the 4 subimages (labelled by a-b-c-d) is suggested, with the appropriate modification of the legend.
Answer. We would like to leave it as it is and already discussed in the text.
Please spell-check the text for typographical errors.
Answer. English proof-reading was made.
Because of the complexity of this review, the repetition of the verbal definitions of the main optical methods, separetely in a Glossary, is suggested.
Answer. We kept abreviations to minimum and they are standard. We rather keep all text shorter.